# Nonequivalent After-Effects of Alternating Current Stimulation on Motor Cortex Oscillation and Inhibition: Simulation and Experimental Study

**DOI:** 10.3390/brainsci12020195

**Published:** 2022-01-31

**Authors:** Makoto Suzuki, Satoshi Tanaka, Jose Gomez-Tames, Takuhiro Okabe, Kilchoon Cho, Naoki Iso, Akimasa Hirata

**Affiliations:** 1Faculty of Health Sciences, Tokyo Kasei University, 2-15-1 Inariyama, Sayama 350-1398, Saitama, Japan; okabe-t@tokyo-kasei.ac.jp (T.O.); cho-k@tokyo-kasei.ac.jp (K.C.); iso-n@tokyo-kasei.ac.jp (N.I.); 2Laboratory of Psychology, Hamamatsu University School of Medicine, 1-20-1 Handayama, Higashi-ku, Hamamatsu 431-3192, Shizuoka, Japan; tanakas@hama-med.ac.jp; 3Department of Electrical and Mechanical Engineering, Nagoya Institute of Technology, Gokiso-cho, Showa-ku, Nagoya 466-8555, Aichi, Japan; jgomez@nitech.ac.jp (J.G.-T.); ahirata@nitech.ac.jp (A.H.); 4Center of Biomedical Physics and Information Technology, Nagoya Institute of Technology, Gokiso-cho, Showa-ku, Nagoya 466-8555, Aichi, Japan

**Keywords:** electric field simulation, oscillation, primary motor cortex, spike-timing-dependent plasticity, transcranial alternating current stimulation

## Abstract

The effects of transcranial alternating current stimulation (tACS) frequency on brain oscillations and cortical excitability are still controversial. Therefore, this study investigated how different tACS frequencies differentially modulate cortical oscillation and inhibition. To do so, we first determined the optimal positioning of tACS electrodes through an electric field simulation constructed from magnetic resonance images. Seven electrode configurations were tested on the electric field of the precentral gyrus (hand motor area). We determined that the Cz-CP1 configuration was optimal, as it resulted in higher electric field values and minimized the intra-individual differences in the electric field. Therefore, tACS was delivered to the hand motor area through this arrangement at a fixed frequency of 10 Hz (alpha-tACS) or 20 Hz (beta-tACS) with a peak-to-peak amplitude of 0.6 mA for 20 min. We found that alpha- and beta-tACS resulted in larger alpha and beta oscillations, respectively, compared with the oscillations observed after sham-tACS. In addition, alpha- and beta-tACS decreased the amplitudes of conditioned motor evoked potentials and increased alpha and beta activity, respectively. Correspondingly, alpha- and beta-tACSs enhanced cortical inhibition. These results show that tACS frequency differentially affects motor cortex oscillation and inhibition.

## 1. Introduction

Primary motor cortex (M1) excitability is affected by the dynamic oscillations of the cerebello-thalamo-cortical network. In general, cortical excitability (CE) has been correlated with dynamic network interactions that are reflected by alpha- and beta-band oscillations [1,2,3], which have been associated with inhibitory γ-aminobutyric acid (GABAergic) interneurons in several cortical regions, including M1 [4,5,6]. Therefore, the oscillatory activity in both alpha- and beta-bands in inhibitory GABAergic interneurons modulates M1 excitability [7,8].

Recently, transcranial alternating current stimulation (tACS), which involves the non-invasive delivery of a weak alternating current to the scalp, has been used to modulate cortical oscillatory activity and excitability in a frequency-specific manner [9,10,11,12,13]. Such modulatory effects have been reported to occur not only during but also after stimulation [9,10,11]. A possible mechanism underlying tACS-induced after-effects is spike-timing-dependent plasticity (STDP) [14]. In STDP, the pre- and post-synaptic potentials resulting from the rhythm of electrical stimulation-derived neuronal excitation, together with the intrinsic oscillatory patterns of the potentials themselves, affect the magnitude and direction of synaptic strength, leading to synaptic long-term potentiation (LTP) or long-term depression (LTD) [10,14,15]. Previous studies have not only shown that alpha-tACS increased the alpha-band power of brain oscillations but also have shown that this effect outlasted the end of stimulation by ≥30 min [10,11,12] and that the strength of the entrained oscillatory power was positively correlated with the strength of the after-effects [16,17]. These results imply that the effects of tACS on brain oscillation and excitability may last beyond stimulation.

Several studies have suggested that alpha and beta oscillations are negatively correlated with CE [18,19], and others have reported decreased corticospinal excitability after 15 Hz and 20 Hz tACS [19,20]. However, some studies have reported no such changes in excitability after 10 Hz and 20 Hz tACS [19,21,22], while one study reported increased corticospinal excitability after 20 Hz tACS [23]. Therefore, the after-effects of tACS remain controversial. One of the reasons for these inconsistencies is the inter-individual variability in electric fields, which is attributable to the different brain anatomy of each individual [24]. Electric fields are unevenly distributed on the cortex, and little is known about the optimal electrode sites and configurations for successful tACS. Accordingly, although it is known that the timing between electrical stimulation and intrinsic neuronal oscillation affects synaptic strength, the effects of frequency on cortical excitability during and after tACS are not fully understood.

To elucidate this topic, computational modeling of the head of each participant can be used to guide the optimal placement of electrodes [25,26,27], which could also aid in predicting the effects of stimulation [28]. Regardless, this method has limited application in the clinical setting in that it requires imaging data for each participant with neurological and mental disorders such as Alzheimer’s disease and Parkinson’s disease [29,30,31,32,33]. One promising approach to solve this is to determine the optimal electrode location for a group of participants based on the montage arrangement that delivers the highest intensity with the lowest individual variability [34,35]. The detection of optimal electrode location with highest intensity and lowest variability could be useful for clinical application by mitigating the need for individual imaging data. Therefore, the present study aimed to explore whether optimal tACS electrode montage arrangement could be obtained from individualized head models analyzed at the group level and to examine the effects of alpha- and beta-tACS delivered through such arrangement on cortical excitability. We hypothesized that, if tACS results in a frequency-specific, STDP-mediated strengthening or weakening of neuronal circuits [14,17], then alpha- and beta-band oscillations would change according to tACS frequency. In particular, 10 Hz oscillations (i.e., alpha-band) would be synchronized with the peak phase of 10 Hz and 20 Hz tACSs (Figure 1A). Similarly, 20 Hz oscillations (i.e., beta-band) would be synchronized with the peak phase of 20 Hz tACS, as well as both the peak and trough phases of 10 Hz tACS (Figure 1B). Correspondingly, the magnitude of cortical inhibition would change with the increasing power of cortical oscillation resulting from different tACS frequencies. Exploring how cortical oscillations and inhibition change after alpha- and beta-tACS may contribute to our understanding of tACS-induced organizational processes.

## 2. Materials and Methods

### 2.1. Participants

Our experimental procedures were approved by the Research Ethics Committee of the Tokyo Kasei University (SKE2018-6) and followed the principles of the Declaration of Helsinki. All participants provided written informed consent prior to participation. In addition, all experiments were performed following the “Guidelines for TMS/tES clinical services and research through the COVID-19 pandemic” [36].

This was a single-center, single-blinded, within-participant study. The selection of the sample size was based on a desired statistical power of 80% for the detection of changes in power spectra and motor evoked potential (MEP) amplitudes, with an effect size of 0.30 and a two-sided α-level of 0.05. According to these parameters, G*Power 3.0 [37] yielded a sample size of 16. Therefore, we recruited 16 healthy volunteers without neurological or psychiatric diseases who were not at risk of adverse events from transcranial magnetic stimulation (TMS) [38] and were not taking any medication. Right-handedness was confirmed with the Edinburgh Handedness Inventory [39].

### 2.2. Electric Field Simulation

A volume conductor model of the anatomical human head model was constructed using magnetic resonance imaging (MRI) images from a database of eighteen participants (all healthy males). The generated tACS electric field was conducted for the following scenarios. We considered seven tACS montages based on the International 10–20 system and anatomical structure (Figure 2A). tACS was applied with an intensity of 0.3 mA throughout the rubber sheets (1.8 × 1.8 cm^2^) corresponding to a stimulation phase. We compared the normal component of the electric field values averaged over the precentral knob in the precentral area (Figure 2B) in the standard brain space among all electrode configurations for group-level analysis. Additionally, relative standard deviation (RSD) was used to quantify how much variability of the electric field was present between participants. Appendix A presents the detailed computational model implementation.

### 2.3. Hotspot Detection

Each participant was comfortably seated with their right hand resting on the testing equipment. The skin overlying the right first dorsal interosseous (FDI) muscle was cleaned with alcohol to reduce its electrical resistance, and recording and reference double differential surface electrodes (FAD-DEMG1, 4Assist, Tokyo, Japan) were placed over the muscle. MEPs from the FDI muscle were recorded, amplified by 100, bandpass-filtered at 10–2000 Hz, digitized at 10 kHz with a PowerLab system (ADInstruments, Dunedin, New Zealand), and stored in a solid-state drive.

After placing a tight-fitting cap over the participant’s head, we drew intersecting nasion–inion and interaural lines on the cap with a marker pencil to localize the vertex (Cz) in accordance with the 10–20 International System. Magstim 200^2^ (Magstim, Whitland, UK) stimulators were employed to deliver TMS as a monophasic current waveform via a cable to the scalp surface through a figure-of-eight coil (internal diameter of each wing: 70 mm). To induce current flow in the left brain along the posterior–lateral to anterior–medial direction, we placed the coil tangentially to the scalp and held the handle so that it would point backwards and sideways, at approximately 45° from the midline. As previously described [40,41], we visually detected the optimal coil position to elicit maximum MEPs in the right FDI muscle (“hotspot”) and marked the location with a soft-tipped pen.

### 2.4. tACS

To determine the after-effects of tACS frequency on brain oscillations and cortical excitability, each participant was tested with two active (alpha- and beta-tACS) and one sham condition (Figure 3A) on three different days.

For all procedures, the participants were seated in a comfortable chair with their eyes open in a quiet room. tACS was delivered by a battery-driven current stimulator (DC Stimulator-Plus; NeuroConn, Ilmenau, Germany) through two rubber electrodes (1.8 × 1.8 cm) attached to the participants’ scalp. Using a conductive and adhesive paste (Ten20 Conductive Paste; Weaver and Company, Aurora, CO, USA) and a support bandage, the electrodes were placed above Cz and CP1, respectively, in accordance with the 10–20 International System. The Cz-CP1 montage was selected because, in our simulation, this configuration produced high current densities with low variability in the hand motor area (see Section 3). For active stimulation, tACS was delivered at a fixed frequency of 10 Hz (alpha-tACS) or 20 Hz (beta-tACS) with a peak-to-peak amplitude of 0.6 mA (current density: 0.093 mA/cm^2^) through a current stimulator for 20 min [22,42]. Sham stimulation was performed at a fixed frequency of 15 Hz with the same intensity for 30 s to cause skin sensations such as tingling [43]; no current was delivered for the remaining 19 min and 30 s. The order of conditions was randomized across participants, and all sessions were separated by ≥1 day. Participants were blinded to the condition.

### 2.5. Electroencephalography

Electroencephalographic (EEG) data were obtained before and after tACS. The skin was prepared with alcohol, and five gold-coated active EEG electrodes were placed at the FDI muscle hotspot (HS), and 2.5 cm lateral anterior (LA), medial anterior (MA), lateral posterior (LP), and medial posterior (MP) to the hotspot, respectively (Figure 3B). Electrodes were mounted in an elastic cap by a holder and covered by support bandages. Electrodes were also placed above the right eye and below the left eye to record activities related to eye movements and blinking.

Participants were instructed to keep their eyes open and fixate on a 0.5 cm blue dot on a screen located about 100 cm in front of them. EEG was performed using the Polymate V (Miyuki Giken, Tokyo, Japan), and data were sampled at 1000 Hz and filtered from 0.15 Hz to 200 Hz. Electrode impedance was maintained at ≤10 kΩ. EEG signals were referenced to the averaged recordings of the right and left earlobes.

### 2.6. Cortical Inhibition Recordings

To measure cortical inhibition and evaluate GABA_A_-mediated inhibitory effects, MEPs and short-interval intra-cortical inhibition (SICI) were measured before and after tACS [44,45]. The hotspot’s resting motor threshold (RMT) was defined as the minimum stimulus intensity required to elicit an MEP ≥50 μV in the relaxed FDI muscle in 5 out of 10 consecutive trials. Unconditioned MEPs for the FDI muscle were evoked at the hotspot at 120% of the RMT value. The stimulus intensity for the first conditioning pulse was set at 80% of the RMT value, and the second test pulse was administered suprathreshold at an intensity of 120% that of the RMT. A 2.5 ms interstimulus interval was used to test SICI [44,45]. Twenty trials of both unconditioned MEP and SICI measurements at a frequency of 0.2 Hz were recorded in random order.

### 2.7. Data Analysis

#### 2.7.1. EEG Data Processing

The six EEG datasets (i.e., data collected before and after alpha-, beta-, and sham-tACS) were each split into 180 non-overlapping 1 s epochs. All epochs were visually inspected, and those containing eye blinks or muscle movement artifacts were excluded. After artifact rejection, the fast Fourier transform was applied for frequencies between 0 and 40 Hz (1 Hz resolution) for individual epochs using a Hanning window. After logarithmically transforming and averaging the power values of the five electrodes, frequency bands of interest were selected in the alpha (10 ± 1 Hz) and beta (20 ± 1 Hz) ranges, taking into account the tACS frequencies.

In order to conduct a proper comparison for differences in power spectra between tACS frequency conditions, normality testing using the Kolmogorov–Smirnov test was used. Based on the result of the Kolmogorov–Smirnov testing, either parametric two-way repeated measures analysis of variance (ANOVA) or nonparametric Friedman’s test was used. Additionally, for nonparametric testing, the logarithmically transformed power spectrum without normality distribution was normalized to baseline (i.e., before tACS) according to the following equation:(1)$$NP\left(f,t\right)=\frac{A\left(f,t\right)-R\left(f\right)}{R\left(f\right)},\;$$
where NP denotes the normalized power spectrum, A represents the EEG power spectrum at time t and frequency f (i.e., the power spectrum of 10 and 20 Hz after tACS), and R denotes the mean power spectrum of the baseline period, defined as the 3 min interval before tACS. A large positive value indicates a large increase in the EEG power spectrum from the baseline period [46]. Furthermore, post hoc analysis with parametric Bonferroni correction or the nonparametric Steel–Dwass test was performed to compare differences in power spectra between tACS frequency conditions.

#### 2.7.2. MEP Data Processing

A previous study [47] noted that MEP amplitudes randomly fluctuate between stimuli. Therefore, peak-to-peak MEP amplitudes were evaluated for the existence of outliers through Tukey’s fences, with values more than 1.5 times that of the interquartile range excluded from the datasets [48]. To increase the precision of level and slope estimations of cortical inhibition, the blank cells produced by removing the outliers were then linearly interpolated. Next, time-series analyses were conducted using the Bayesian method. The local linear trend model (LLT) assumes that both the level (Equation (3)) and slope (Equation (4)) of the trend from observational values (Equation (2)) follow Gaussian random walks. LLTs were constructed for the MEP amplitudes as follows:(2)$$y_{t}=\mu_{t}+\varepsilon_{t}\;,\;$$
(3)$$\mu_{t+1}=\mu_{t}+\nu_{t}+\xi_{t}\;,$$
(4)$$\nu_{t+1}=\nu_{t}+\zeta_{t}\;,$$
where $y_{t}$ is the observational value; $\varepsilon_{t}$ indicates random variables; $\mu_{1,}$ represents the initial level; $\nu_{1}$ is the initial slope; and $\xi_{t}$ and $\zeta_{t}$ indicate disturbances in the level and slope, respectively [49].

After state value estimation, normality testing using Kolmogorov–Smirnov test was used. Based on the result of the Kolmogorov–Smirnov testing, either parametric two-way repeated measures ANOVA or nonparametric Friedman’s test was used. For nonparametric testing, conditioned and unconditioned MEP amplitudes were normalized to the baseline data (Equation (1)). In Equation (1), NP denotes normalized MEPs, A denotes MEPs at time t, and R denotes the mean MEP of the baseline period before tACS. A great positive value indicates a large increase in MEPs compared with that in the baseline period [46]. Post hoc analysis with parametric Bonferroni correction or the nonparametric Steel–Dwass test was performed to compare differences in MEP amplitudes among the three tACS conditions.

Data analysis was conducted with EMSE (Miyuki Giken, Tokyo, Japan), the SciPy package in the Python environment (Python Software Foundation, Wilmington, DE, USA), and the R 3.4.0 software (The R Foundation, Vienna, Austria). Data are expressed as means ± standard errors of the mean (SEM). Statistical significance was set at *p* < 0.05.

## 3. Results

### 3.1. Electric Felds of Cortical tACS

Figure 4 shows the group-level electric field distribution (normal component) on standard cortical brain space for different montages. The electric field was induced in the precentral gyrus for tACS with a 3.24 cm^2^ rubber sheet and an intensity of 0.3 mA on each phase. However, the field focality was not identical between the montages. In addition, we compared the averaged electric field values in the hand knob among the montages (Table 1 and Figure 4). The higher averaged values corresponded to Cz-CP1 as 0.12 V/m (min = 0.04 V/m, max = 0.22 V/m). Additionally, we found less variability in the induced electric field among participants for the Cz-CP1 (±standard deviation (SD) = 0.05 V/m, relative SD = 38%) and FC1-Pz (±SD = 0.04 V/m, relative SD = 40%), as shown in Table 1 and Figure 4. In summary, the mean of the normal component averaged over the hand knob was higher and more stable by selecting the Cz-CP1 montage at group-level analysis, and at the same time the Cz-CP1 montage of tACS not interfering with the EEG recording for the M1 region.

### 3.2. Changes in Brain Oscillation and Excitation

A total of 4 men and 12 women aged 20–40 years (25.1 ± 7.5 years) were enrolled, and the mean laterality quotient score was 0.9 (SD = 0.1). Figure 5 shows the power spectrum grand-averaged across all participants. As shown in Figure 5, the power spectrum of alpha-band oscillations was increased after alpha-tACS, whereas that of beta-band oscillations was increased after beta-tACS. However, sham-tACS did not result in any changes in either power spectrum.

The Kolmogorov–Smirnov test showed that the power spectra lacked normality (alpha oscillation before and after alpha-tACS, beta-tACS, and sham-tACS: both *p* < 0.0001). Therefore, nonparametric testing and Equation (1) was used for comparison of the power spectra of alpha and beta oscillations after alpha-, beta-, and sham-tACS treatments. The normalized power changes (i.e., the event-related synchronization (ERS) and event-related desynchronization (ERD)) in alpha-band oscillatory neural activities after alpha-, beta-, and sham-tACSs are shown in Figure 6A. The Friedman test showed a significant difference in power changes in alpha-band oscillations among alpha-, beta-, and sham-tACSs (chi-squared = 16.75, degree of freedom = 2, *p* = 0.0002). Additionally, post hoc tests showed that alpha power oscillation was greater after alpha-tACS than after sham-tACS (alpha-tACS vs. beta-tACS: t = 1.37, *p* = 0.358; alpha-tACS vs. sham-tACS: t = 3.51, *p* = 0.001; beta-tACS vs. sham-tACS: t = 2.29, *p* = 0.057). Alpha power oscillation was greater after beta-tACS than after sham-tACS, but significance was not reached.

The ERS/ERD of beta-oscillatory neural activities after alpha-, beta-, and sham-tACSs are shown in Figure 6B. The Friedman test showed a significant difference in the ERS/ERD of beta oscillations among the alpha-, beta-, and sham-tACSs (chi-squared = 11.53, degree of freedom = 2, *p* = 0.003). Additionally, post hoc tests showed that beta power oscillation was greater after beta-tACS than after sham-tACS (alpha-tACS vs. beta-tACS: t = 1.65, *p* = 0.358; alpha-tACS vs. sham-tACS: t = 3.45, *p* < 0.0001; beta-tACS vs. sham-tACS: t = 4.34, *p* = 0.016). Moreover, beta power oscillation was lower after alpha-tACS than after sham-tACS.

The grand-averaged actual and estimated peak-to-peak MEP amplitudes according to the LLT are shown in Figure 7. The actual MEP amplitudes fluctuated randomly before and after alpha-, beta-, and sham-tACS, whereas the fluctuation in the estimated MEP amplitudes was reduced by the LLT.

The Kolmogorov–Smirnov test showed that the MEP amplitudes lacked normality (conditioned and unconditioned MEP before and after alpha-tACS, beta-tACS, and sham-tACS: both *p* < 0.0001). Therefore, nonparametric testing and Equation (1) was used for comparison of the conditioned and unconditioned MEP amplitudes after alpha-, beta-, and sham-tACS treatments. The changes in the normalized conditioned MEP amplitudes after tACS are shown in Figure 8A. The Friedman test showed a significant difference in normalized condition MEP amplitudes among alpha-, beta-, and sham-tACSs (chi-squared = 42.28, degree of freedom = 2, *p* < 0.0001). Further, post hoc tests showed that the conditioned MEP amplitudes were smaller after alpha- and beta-tACS than after sham-tACS. Specifically, MEP amplitudes after alpha-tACS were smaller than those after beta-tACS (alpha-tACS vs. beta-tACS: t = 2.56, *p* = 0.029; alpha-tACS vs. sham-tACS: t = 4.93, *p* < 0.0001; beta-tACS vs. sham-tACS: t = 2.38, *p* = 0.045).

The normalized unconditioned MEP amplitudes changes after tACS are shown in Figure 8B. The Friedman test showed a significant difference among the alpha-, beta-, and sham-tACSs (chi-squared = 11.93, degree of freedom = 2, *p* = 0.002). Further, post hoc tests showed that the unconditioned MEP amplitude was smaller after alpha- and beta-tACS than after sham-tACS (alpha-tACS vs. beta-tACS: t = 1.41, *p* = 0.338; alpha-tACS vs. sham-tACS: t = 3.80, *p* = 0.0004; beta-tACS vs. sham-tACS: t = 2.57, *p* = 0.028).

## 4. Discussion

Previous studies did not conduct a group-level evaluation of the optimal sites and montage configurations for tACS electrodes [7,13,19,20,21,22,23,42]. Therefore, we investigated this through a computational simulation, which showed that the tACS Cz-CP1 montage arrangement diminishes the inter-individual variability in the electric field. The electric field range for this arrangement (0.1  V/m–0.2  V/m) indicated a modulatory effect [50]. Therefore, we utilized the Cz-CP1 montage configuration to deliver tACS. We opted for a group-level electric field analysis to maximize the electric field on the target while minimizing individual variability to determine the optimal montage applied to all participants (one-for-all) based on the International 10–20 system positioning. This is advantageous with respect to the individual-level electric field analysis, as it does not require imaging of each individual or electrode localization based on a navigation system that is not always available in clinical settings, and it did not increase the participants’ time in the experiment, which was limited during the COVID-19 pandemic [36]. Therefore, our simulation based on group-level electric field analysis is advantageous for adaptation to various clinical settings, obviating the need for imaging data in the individual-level electric field analysis.

Our experimental results show that alpha- and beta-tACS result in larger alpha and beta oscillations, respectively, and they differently influence cortical inhibition. In addition, alpha-, beta-, and sham-tACS result in a stepwise decrease in conditioned MEP amplitudes. These observations show that alpha- and beta-tACS differently modulate alpha- and beta-band oscillations, which, in turn, differently influence cortical inhibition. This implies that cortical oscillation and inhibition are not equally affected by alpha- and beta-tACSs. Previous studies have noted that tACS has online and offline modulatory effects during and after stimulation [9,10,11]. Especially, the offline effects after tACS underline its potential as a therapeutic tool because of its lasting effect beyond stimulation period.

Two possible mechanisms of tACS modulatory effect have been suggested. First, tACS directly entrains intrinsic brain oscillations [51,52,53,54]. Second, tACS leads to synaptic changes via STDP mechanisms [10,14,15,17]. In entrainment of brain oscillation, intrinsic brain oscillation in accordance with the external stimulation frequency will be entrained but intrinsic brain oscillation outside the stimulation frequency will not be affected [11,55]. Therefore, intrinsic alpha and beta oscillation in M1 [56,57] are externally tuned by tACS, according to the resonance-like hypothesis [23]. Adding to the entrainment mechanism, STDP could possibly explain tACS-induced after-effects [14]. Synapses are either strengthened or weakened depending on the timing of their input and output activity [10,14,15,17], which might be related to the after-effects of tACS [58]. According to the STDP model, facilitatory effects on neural oscillations are expected with the synchronization between the peak phase of tACS and the up state of neural firing, whereas depressive effects are expected with the asynchronization between the two events (i.e., the trough phase of tACS synchronizing the up state of neural firing) [10,14,15,17,55,58]. In agreement with these predictions, we observed that alpha- and beta-tACS increased alpha and beta oscillatory activity, respectively, while alpha-tACS decreased beta oscillatory activity. One possible explanation for this decrement could be that both the peak and trough phases of alpha-tACS may synchronize the up state of neural firing. Previous studies have already showed an enhanced oscillatory activity after tACS [10,14,15,17,58], but we found that tACS frequency influences brain oscillatory activity in a frequency-dependent manner.

Kiers et al. [47] noted that MEP amplitudes randomly fluctuated during the recording period, and Ogata et al. [18] suggested that, during the resting state, the relationship between amplitude fluctuations and cortical oscillations and inhibition is not conclusive. To solve these issues, we used the LLT model based on Bayesian estimations to examine the effects of alpha- and beta-tACS on the level and slope of cortical inhibition, thus eliminating the confounding factor of amplitude fluctuations during the resting state.

Alpha- and beta-band oscillations have been associated with the inhibitory GABAergic interneurons within M1 [4,5,6,59]. Accordingly, enhanced GABAergic interneuron activity plays an essential role in the modulation of M1 excitability induced by alpha- and beta-tACS [60]. However, a previous study found no effect of beta-tACS in SICI after TMS-induced GABA_A_ inhibition [7,45]. Our findings on the alpha- and beta-tACSs effects on SICI suggest that the cortex is inhibited after stimulation. One possible explanation for this discrepancy could be that changes in the M1 post-synaptic potentials that correspond to alpha- and beta-oscillations can be altered by alpha- and beta-tACS, assuming that tACS enhances the synaptic strength of GABAergic neurons, as described by the STDP model [10,14,15,17,58]. Therefore, cortical inhibition might be affected by the enhancement of cortical oscillations associated with tACS frequency.

Previous work has shown that beta oscillations correlate with CE levels [61,62,63] and that alpha and beta oscillations inhibited MEP amplitudes [18,64,65,66], although a statistically significant relationship has not always been observed [67,68]. In addition, several studies noted that 20 Hz tACS increased corticospinal excitability [13,23]. However, Cappon et al. [69] and Zaghi et al. [20] observed reduced MEP amplitudes after 15 and 20 Hz tACS of M1, and Wach et al. [21] and Schutter and Hortensius [19] did not find any effect of 10 and 20 Hz tACS. However, in our study, unconditioned MEP amplitudes decreased after alpha- and beta-tACSs. Sanger et al. [70] suggested that MEP amplitude was affected not only by GABA_A_ receptors, thus suggesting SICI, but also by excitatory glutamatergic and inhibitory GABA_B_ receptors. However, the precise mechanism of tACS-related changes in unconditioned MEP amplitude is still unclear. Future studies should investigate changes in CE and inhibition related to GABA_A_, GABA_B_, and glutamatergic receptors at various tACS frequencies.

Previous studies indicate that neurological and mental diseases induce changes in brain oscillations [29,30,31,32,33,71,72]. In Parkinson’s disease, abnormal beta activity could be related to bradykinesia [73]. Additionally, in Alzheimer’s disease, abnormal alpha activity could be related to memory dysfunction [74]. The frequency-specific tACS modulatory effects for brain oscillation and inhibition could have potentially useful clinical applications. Further studies are needed to assess the tACS modulatory effect on brain oscillatory and inhibitory disorders in neurological and mental disorders such as Parkinson’s disease and Alzheimer’s disease.

Our study has two main limitations. First, because it was necessary to monitor tACS waves for safe and precise stimulation, we could not utilize a double-blinded design. Therefore, double-blinded studies should be conducted in the future. Second, the sample size and composition, which was estimated using G*Power 3.0 [24], were limited as we did not consider factors such as differences in age, sex, baseline MEP sizes, or MEP’s latency. In fact, concerning these two latter factors, Wiethoff et al. [75] noted that baseline MEP sizes and latency differences in MEP (anteroposterior stimulation minus latero-medial stimulation) result in changes in corticospinal excitability. Thus, future studies need to include a larger sample size to analyze the effects of such parameters on cortical excitability.

## 5. Conclusions

In conclusion, we found that changes in tACS frequency result in corresponding changes in alpha- and beta-band oscillations and cortical inhibition. These results imply that cortical oscillations can be differentially altered by tACS and that cortical inhibition may change according to the tACS frequency-modulated balance between alpha and beta oscillations.

## Figures and Tables

**Figure 1 brainsci-12-00195-f001:**
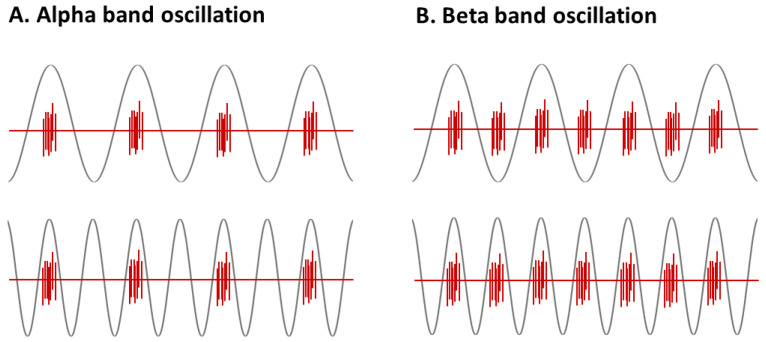
The hypothesized relationship between tACS frequency and neuronal activity. Gray lines denote 10 Hz (top trace) and 20 Hz (bottom trace) tACS, and red lines denote up and down states of neural firing. We hypothesized that (**A**) 10 Hz oscillations would be synchronized with the peak phase of 10 Hz and 20 Hz tACS, and (**B**) 20 Hz oscillations would be synchronized with both the peak and trough phases of 10 Hz tACS, as well as the peak phase of 20 Hz tACS.

**Figure 2 brainsci-12-00195-f002:**
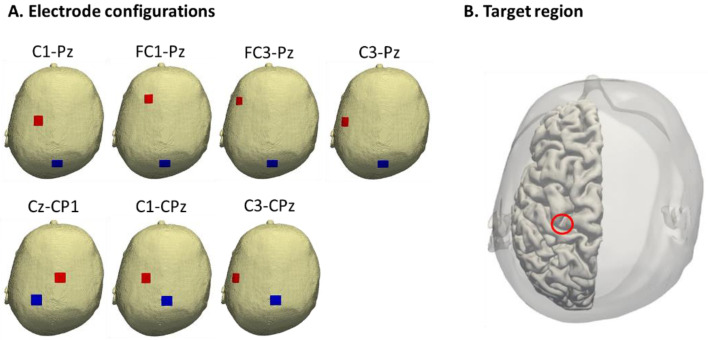
Electric field simulation. Seven tACS electrode configurations based on the International 10–20 system (C1-Pz, FC1-Pz, FC3-Pz, C3-Pz, Cz-CP1, C1-CPz, and C3-CPz) and anatomical structure from MRI image (**A**). The electric fields induced by each of the seven tACS montages were averaged on the precentral knob in the precentral area (**B**).

**Figure 3 brainsci-12-00195-f003:**
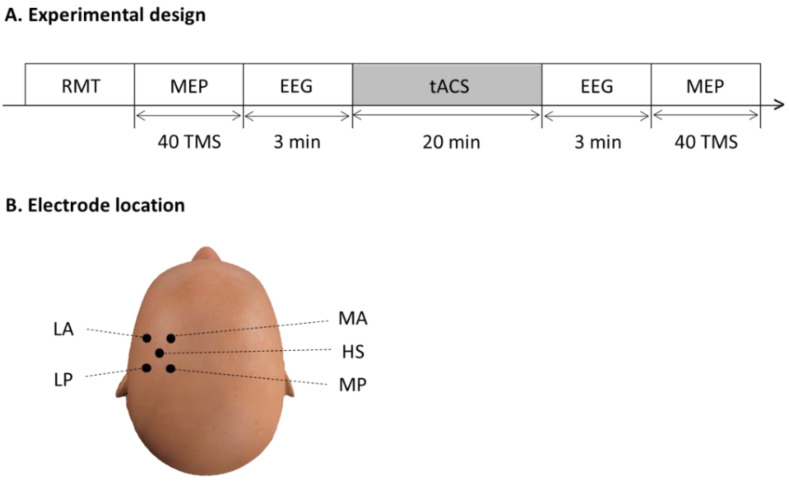
(**A**) The experimental design for testing the effects of the alpha- (10 Hz), beta- (20 Hz), and sham-tACS conditions on brain oscillations and cortical inhibition. Testing was performed on three different days. (**B**) Five EEG electrodes were placed at the FDI muscle hotspot (HS), and 2.5 cm lateral anterior (LA), medial anterior (MA), lateral posterior (LP), and medial posterior (MP) to the hotspot.

**Figure 4 brainsci-12-00195-f004:**
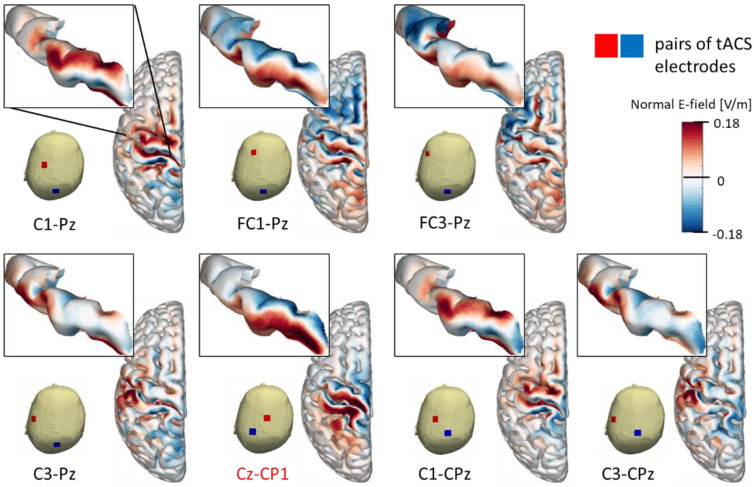
Normal component of the electric field (group-level analysis, *n* = 18) during tACS in seven montages. For practical comparison, the tACS phase depicted here was chosen so that the electric field’s normal component is towards the precentral wall.

**Figure 5 brainsci-12-00195-f005:**
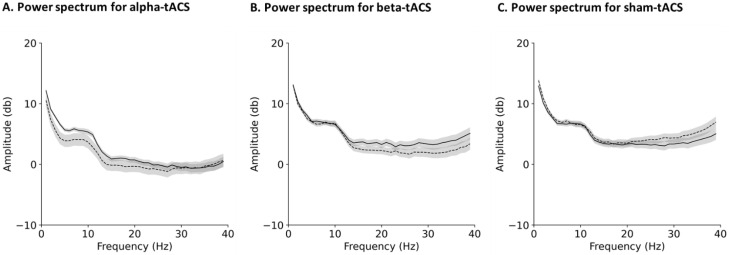
Grand-averaged power spectra before and after (**A**) alpha-, (**B**) beta-, and (**C**) sham-tACSs. Dashed and solid lines denote the power spectra before and after tACS, respectively. Shaded areas indicate the standard error of the mean. The power spectrum of alpha-band oscillation was increased after alpha-tACS, whereas beta-band oscillation was increased after beta-tACS. However, the power spectra of alpha- and beta-band oscillations were not changed by sham-tACS.

**Figure 6 brainsci-12-00195-f006:**
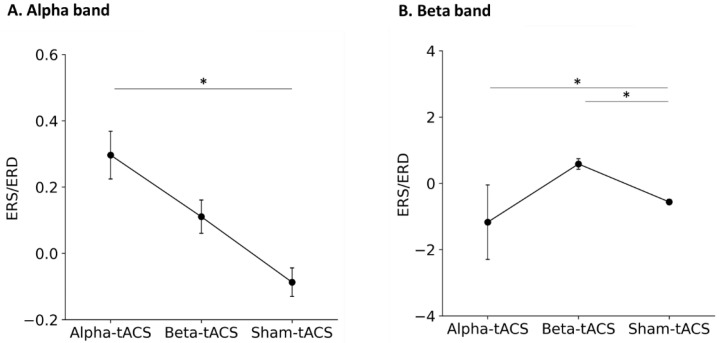
Normalized power changes in (**A**) alpha and (**B**) beta-oscillatory neural activity after alpha-, beta-, and sham- tACSs. Dots and error bars denote the mean and standard error of the mean, respectively. Alpha-tACS resulted in an increase in alpha power oscillations and decreased beta power oscillations, whereas beta-tACS increased beta power oscillations. *: *p* < 0.05.

**Figure 7 brainsci-12-00195-f007:**
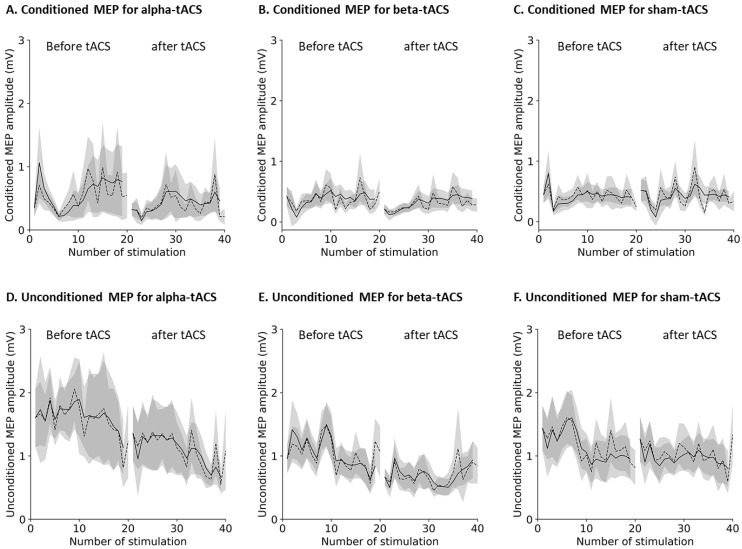
Grand-averaged time-series of the (**A**–**C**) conditioned and (**D**–**F**) unconditioned MEP amplitudes by the LLT model. Dashed and solid lines indicate actual and estimated MEP amplitudes, respectively. Shaded areas indicate the standard error of the mean. The actual MEP amplitudes fluctuated randomly, whereas the fluctuation of estimated MEP amplitudes was reduced by the LLT model.

**Figure 8 brainsci-12-00195-f008:**
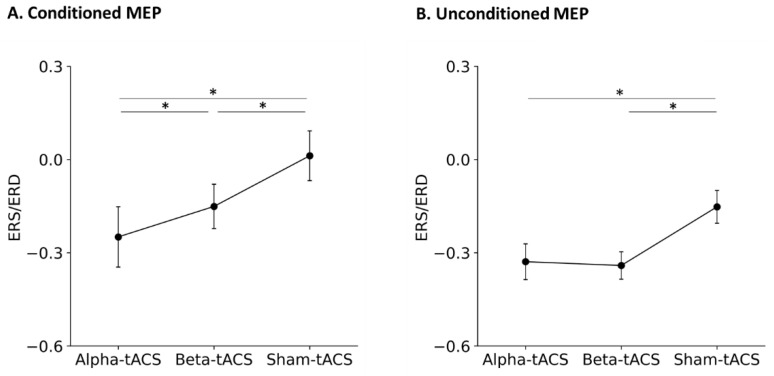
The normalized (**A**) conditioned and (**B**) unconditioned MEP amplitude among alpha-, beta-, and sham-tACS. Dots and error bars denote the mean and standard error of the mean, respectively. Alpha- and beta-tACSs decreased both conditioned and unconditioned MEP amplitudes. *: *p* < 0.05.

**Table 1 brainsci-12-00195-t001:** Normal component of electric field (group-level) averaged over the precentral knob.

tACS Montage	Group-Level (mV/m)	RSD (%)
C1-Pz	79	141
FC1-Pz	107	40
FC3-Pz	74	77
C3-Pz	58	56
Cz-CP1	120	38
C1-CPz	65	168
C3-CPz	59	58

RSD, relative standard deviation; tACS, transcranial alternating current stimulation.

## Data Availability

Raw data were generated at Tokyo Kasei University. Derived data supporting the findings of this study are available from the corresponding author, M.S.

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
