# Peer review of "Nonequivalent After-Effects of Alternating Current Stimulation on Motor Cortex Oscillation and Inhibition: Simulation and Experimental Study"

_brainsci, 2022, doi:10.3390/brainsci12020195_

Round 1

Reviewer 1 Report

Suzuki et al. have studied the effect of tACS on brain oscillation in a frequency dependant manner. They have noted that changes in tACS frequency result in corresponding changes in alpha and beta band oscillations and cortical inhibition. They have also tried to find out the optimal montage (Cz-CP1) for delivering tACS. I have some suggestions:

  1. Overall, the manuscript is well written and has implication in the clinical field of movement disorders. However, that clinical correlation should be highlighted. tACS has already been tried in Parkinson’s disease, tremor and cervical dystonia.

(Ref: Ganguly J, Murgai A, Sharma S, Aur D, Jog M. Non-invasive Transcranial Electrical Stimulation in Movement Disorders. Front Neurosci. 2020 Jun 5;14:522. doi: 10.3389/fnins.2020.00522)

  1. Results of the clinical studies with tACS done so far were varied and one of the main reasons for that was heterogenous montage selection. This manuscript should point out that and mention the clinical importance of their proposed best montage.
  2. A brief paragraph on the pathological brain oscillation in at least Parkinson’s disease should be included. The importance of frequency dependent modulation of those pathological oscillation must be mentioned there.

The overall message of the manuscript is clear and the study is well conceptualized. The addition of its clinical implication is missing and should be precisely added to highlight the importance of this nice study and also to make it more interesting.

Author Response

Manuscript ID: brainsci-1543844

Title: Nonequivalent after-effects of alternating current stimulation on motor cortex oscillation and inhibition: simulation and experimental study

Point-by-Point Response

Thank you very much for your favorable comments on our study and your helpful suggestions. We have revised our manuscript after fully considering its contents and by referring to the reviewers’ useful comments and suggestions. Our responses to your comments are as presented below. Please note that the changes made do not influence the content, conclusions, or framework of the paper. We have not listed below all minor changes made; however, these are underlined in the revised manuscript.

Responses to Reviewer 1

Comment 1

Overall, the manuscript is well written and has implication in the clinical field of movement disorders. However, that clinical correlation should be highlighted. tACS has already been tried in Parkinson’s disease, tremor and cervical dystonia. (Ref: Ganguly J, Murgai A, Sharma S, Aur D, Jog M. Non-invasive Transcranial Electrical Stimulation in MovementDisorders. Front Neurosci. 2020 Jun 5;14:522. doi:10.3389/fnins.2020.00522)

Response to comment 1

As you noted, clinical correlation is important for clinical generalizability of our study. We have therefore added the following text to the Discussion (page 12, lines 433–440):

“Previous studies indicate that neurological and mental diseases induce changes in brain oscillations [29-33,71,72]. In Parkinson’s disease, abnormal beta activity could be related to motor bradykinesia [73]. Additionally, in Alzheimer’s disease, abnormal alpha activity could be related to memory dysfunction [74]. The frequency-specific tACS modulatory effects for brain oscillation and inhibition could have potentially useful clinical applications. Further studies are needed to assess the tACS modulatory effect on brain oscillatory and inhibitory disorders in neurological and mental disorders such as Parkinson’s disease and Alzheimer’s disease.”

We have also added the following references:

  1. Brittain, J.S.; Probert-Smith, P.; Aziz, T.Z.; Brown, P. Tremor suppression by rhythmic transcranial current stimulation. Curr Biol 2013, 23, 436-440.
  2. Uhlhaas, P.J.; Singer, W. Neural synchrony in brain disorders: relevance for cognitive dysfunctions and pathophysiology. Neuron 2006, 52, 155-168.
  3. Uhlhaas, P.J.; Singer, W. Neuronal dynamics and neuropsychiatric disorders: toward a translational paradigm for dysfunctional large-scale networks. Neuron 2012, 75, 963-980.
  4. Latreille, V.; Carrier, J.; Gaudet-Fex, B.; Rodrigues-Brazete, J.; Panisset, M.; Chouinard, S.; Postuma, R.B.; Gagnon, J.F. Electroencephalographic prodromal markers of dementia across conscious states in Parkinson's disease. Brain 2016, 139, 1189-1199.
  5. Cozac, V.V.; Gschwandtner, U.; Hatz, F.; Hardmeier, M.; Ruegg, S.; Fuhr, P. Quantitative EEG and Cognitive Decline in Parkinson's Disease. Parkinsons Dis 2016, 2016, 9060649.
  6. .Schnitzler, A.; Gross, J. Normal and pathological oscillatory communication in the brain. Nat Rev Neurosci 2005, 6, 285-296.
  7. Ganguly, J.; Murgai, A.; Sharma, S.; Aur, D.; Jog, M. Non-invasive Transcranial Electrical Stimulation in Movement Disorders. Front Neurosci 2020, 14, 522.
  8. Hammond, C.; Bergman, H.; Brown, P. Pathological synchronization in Parkinson's disease: networks, models and treatments. Trends Neurosci 2007, 30, 357-364.
  9. Schmidt, M.T.; Kanda, P.A.; Basile, L.F.; da Silva Lopes, H.F.; Baratho, R.; Demario, J.L.; Jorge, M.S.; Nardi, A.E.; Machado, S.; Ianof, J.N.; et al. Index of alpha/theta ratio of the electroencephalogram: a new marker for Alzheimer's disease. Front Aging Neurosci 2013, 5, 60.

Comment 2

Results of the clinical studies with tACS done so far were varied and one of the main reasons for that was heterogenous montage selection. This manuscript should point out that and mention the clinical importance of their proposed best montage.

Response to comment 2

We appreciate your helpful comment and have revised the Introduction (page 2, lines 70–77) and Discussion (page 11, lines 370–372) as follows:

“To elucidate this topic, computational modeling of the head of each participant can be used to guide the optimal placement of electrodes, [25-27] which could also aid in predicting the effects of stimulation [28]. Regardless, this method has limited application in the clinical setting in that it requires imaging data for each participant with neurological and mental disorders such as Alzheimer’s disease and Parkinson’s disease [29-33]. One promising approach to solve this is to determine the optimal electrode location for a group of participants based on the montage arrangement that delivers the highest intensity with the lowest individual variability [34,35]. The detection of optimal electrode location with highest intensity and lowest variability could be useful for clinical application by mitigating the need for individual imaging data. Therefore, the present study aimed to explore whether optimal tACS electrode montage arrangement could be obtained from individualized head models analyzed at the group level, and to examine the effects of alpha- and beta-tACS delivered through such arrangement on cortical excitability.”

“This is advantageous with respect to the individual-level electric field analysis as it does not require imaging of each individual or electrode localization based on a navigation system that is not always available in clinical settings, and did not increase the participants’ time in the experiment, which was limited during the COVID-19 pandemic [36]. Therefore, our simulation based on group-level electric field analysis is advantageous for adaptation to various clinical settings, obviating the need for imaging data in the individual-level electric field analysis.”

Comment 3

A brief paragraph on the pathological brain oscillation in at least Parkinson’s disease should be included. The importance of frequency dependent modulation of those pathological oscillation must be mentioned there.

Response to comment 3

Related to Comment 1, we have added the following text to Discussion (page 12, lines 433–440):

“Previous studies indicate that neurological and mental diseases induce changes in brain oscillations [29-33,71,72]. In Parkinson’s disease, abnormal beta activity could be related to motor bradykinesia [73]. Additionally, in Alzheimer’s disease, abnormal alpha activity could be related to memory dysfunction [74]. The frequency-specific tACS modulatory effects for brain oscillation and inhibition could have potentially useful clinical applications. Further studies are needed to assess the tACS modulatory effect on brain oscillatory and inhibitory disorders in neurological and mental disorders such as Parkinson’s disease and Alzheimer’s disease.”

We have also added the following references:

  1. Brittain, J.S.; Probert-Smith, P.; Aziz, T.Z.; Brown, P. Tremor suppression by rhythmic transcranial current stimulation. Curr Biol 2013, 23, 436-440.
  2. Uhlhaas, P.J.; Singer, W. Neural synchrony in brain disorders: relevance for cognitive dysfunctions and pathophysiology. Neuron 2006, 52, 155-168.
  3. Uhlhaas, P.J.; Singer, W. Neuronal dynamics and neuropsychiatric disorders: toward a translational paradigm for dysfunctional large-scale networks. Neuron 2012, 75, 963-980.
  4. Latreille, V.; Carrier, J.; Gaudet-Fex, B.; Rodrigues-Brazete, J.; Panisset, M.; Chouinard, S.; Postuma, R.B.; Gagnon, J.F. Electroencephalographic prodromal markers of dementia across conscious states in Parkinson's disease. Brain 2016, 139, 1189-1199.
  5. Cozac, V.V.; Gschwandtner, U.; Hatz, F.; Hardmeier, M.; Ruegg, S.; Fuhr, P. Quantitative EEG and Cognitive Decline in Parkinson's Disease. Parkinsons Dis 2016, 2016, 9060649.
  6. .Schnitzler, A.; Gross, J. Normal and pathological oscillatory communication in the brain. Nat Rev Neurosci 2005, 6, 285-296.
  7. Ganguly, J.; Murgai, A.; Sharma, S.; Aur, D.; Jog, M. Non-invasive Transcranial Electrical Stimulation in Movement Disorders. Front Neurosci 2020, 14, 522.
  8. Hammond, C.; Bergman, H.; Brown, P. Pathological synchronization in Parkinson's disease: networks, models and treatments. Trends Neurosci 2007, 30, 357-364.
  9. Schmidt, M.T.; Kanda, P.A.; Basile, L.F.; da Silva Lopes, H.F.; Baratho, R.; Demario, J.L.; Jorge, M.S.; Nardi, A.E.; Machado, S.; Ianof, J.N.; et al. Index of alpha/theta ratio of the electroencephalogram: a new marker for Alzheimer's disease. Front Aging Neurosci 2013, 5, 60.

Reviewer 2 Report

This is an interesting, well-designed, high-quality study about the effects of tACS on Motor cortex brain oscillations and cortical inhibition.

The Authors correctly notice the lack of studies about the effects of tACS on cortical excitability during and after the stimulation. To investigate this topic, they firstly modelled a computational model of the brain in a group of (eighteen) subjects based on the electrodes setting that delivers the highest intensity with the lower individual variability, and then collected data from sixteen subjects both using EEG (cortical excitability) and MEP (cortical inhibition). The aim of the study was to investigate whether alpha and beta brain oscillations and cortical inhibition would change according to tACS alpha and beta frequency. The results show that alpha and beta tACS modulated alpha and beta bands, respectively, and that both stimulation protocols decreased conditioned MEP. Authors conclude that alpha and beta tACS differently affect motor cortex oscillations and inhibition.

Overall, the topic investigated is innovative and the study is well-designed, and I agree with the Authors that the impact of electrical stimulation on cortical excitability is poorly investigated in literature. Moreover, I think that more studies should address this topic employing EEG and MEP, as the Authors did. This contribution is surely suited for the publication in Brain Sciences.

Few comments should however be addressed:

- If I am not wrong, no task has been administered during tACS stimulation. Some studies reported that the inclusion of a task during the stimulation might enhance the activation induced by the stimulation itself (e.g., see Miniussi et al., 2013, Neuroscience and Biobehavioral Reviews, https://doi.org/10.1016/j.neubiorev.2013.06.014). Authors should explain why they chose an offline stimulation.

- It was proposed that tACS might be more effective when administered in phase with the natural cortical oscillations (e.g., see Guerra et al., 2016, Cerebral Cortex, https://doi.org/10.1093/cercor/bhw245). Authors should discuss this point.

- EEG data were normalized to the baseline. To me, it is not clear the reason why of this data transformation. Are the data non-normally distributed? I suggest reporting the statistics proving that. Moreover, they could choose other ways to control for non-normality of data, e.g., simply calculating the natural logarithm. Authors should elucidate this point.

- Again, I am confused why they applied a nonparametric test like the Friedman test. In the case of non-normality, they could transform data and then submit the values to an analysis of variance with time as within-subjects factor and evaluate whether there was a main effect (e.g., a significant change between pre and post stimulation). Authors should better motivate their statistical choice . Also, in the results obtained with the Friedman test, I suggest reporting Chi squares and degrees of freedom.

- In the Discussion section, page 10, line 15, Authors say: “In addition, beta-tACS increases alpha oscillations as well”. This is not true based on the results reported, as the difference in alpha power after beta-tACS and after sham-tACS did not reach significance (p = 0.057). Here I suggest removing this sentence and limiting the discussion to the main findings, e.g., alpha- and beta-tACS modulated alpha and beta oscillations, respectively, and they differently influence cortical inhibition.

After having addressed these points, I think this study could deserve the publication in Brain Sciences.

Author Response

Manuscript ID: brainsci-1543844

Title: Nonequivalent after-effects of alternating current stimulation on motor cortex oscillation and inhibition: simulation and experimental study

Point-by-Point Response

Thank you very much for your favorable comments on our study and your helpful suggestions. We have revised our manuscript after fully considering its contents and by referring to the reviewers’ useful comments and suggestions. Our responses to your comments are as presented below. Please note that the changes made do not influence the content, conclusions, or framework of the paper. We have not listed below all minor changes made; however, these are underlined in the revised manuscript.

Responses to Reviewer 2

Comment 1

If I am not wrong, no task has been administered during tACS stimulation. Some studies reported that the inclusion of a task during the stimulation might enhance the activation induced by the stimulation itself (e.g., see Miniussi et al., 2013, Neuroscience and Biobehavioral Reviews, https://doi.org/10.1016/j.neubiorev.2013.06.014). Authors should explain why they chose an offline stimulation.

Response to comment 1

As you noted, it is important to explain the reason for choosing offline stimulation. We have therefore revised the Discussion (page 11, lines 379–382) as follows:

“Previous studies have noted that tACS has online and offline modulatory effects during and after stimulation [9-11]. Especially, the offline effects after tACS underline its potential as a therapeutic tool because of its lasting effect beyond stimulation period.”

Comment 2

It was proposed that tACS might be more effective when administered in phase with the natural cortical oscillations (e.g.,see Guerra et al., 2016, Cerebral Cortex, https://doi.org/10.1093/cercor/bhw245). Authors should discuss this point.

Response to comment 2

We have revised the Discussion (page 11, lines 383–390) as follows:

“Two possible mechanisms of tACS modulatory effect have been suggested. Firstly, tACS directly entrains intrinsic brain oscillations [51-54]. Secondly, tACS leads to synaptic changes via STDP mechanisms [10,14,15,17]. In entrainment of brain oscillation, intrinsic brain oscillation in accordance with the external stimulation frequency will be entrained but intrinsic brain oscillation outside the stimulation frequency will not be affected [11,55]. Therefore, intrinsic alpha and beta oscillation in M1 [56,57] are externally tuned by tACS, according to the resonance-like hypothesis [23]. Adding to the entrainment mechanism, STDP could possibly explain tACS-induced after-effects [14].”

We have also added the following reference:

  1. Herrmann, C.S.; Rach, S.; Neuling, T.; Struber, D. Transcranial alternating current stimulation: a review of the underlying mechanisms and modulation of cognitive processes. Front Hum Neurosci 2013, 7, 279.
  2. Antal, A.; Paulus, W. Transcranial alternating current stimulation (tACS). Front Hum Neurosci 2013, 7, 317.
  3. Reato, D.; Rahman, A.; Bikson, M.; Parra, L.C. Effects of weak transcranial alternating current stimulation on brain activity-a review of known mechanisms from animal studies. Front Hum Neurosci 2013, 7, 687.
  4. Thut, G.; Miniussi, C. New insights into rhythmic brain activity from TMS-EEG studies. Trends Cogn Sci 2009, 13, 182-189.
  5. Frohlich, F.; McCormick, D.A. Endogenous electric fields may guide neocortical network activity. Neuron 2010, 67, 129-143.
  6. Wetmore, D.Z.; Baker, S.N. Post-spike distance-to-threshold trajectories of neurones in monkey motor cortex. J Physiol 2004, 555, 831-850.
  7. Chen, D.; Fetz, E.E. Characteristic membrane potential trajectories in primate sensorimotor cortex neurons recorded in vivo. J Neurophysiol 2005, 94, 2713-2725.

Comment 3

EEG data were normalized to the baseline. To me, it is not clear the reason why of this data transformation. Are the data non-normally distributed? I suggest reporting the statistics proving that. Moreover, they could choose other ways to control for non-normality of data, e.g., simply calculating the natural logarithm. Authors should elucidate this point.

Response to comment 3

As you noted, we should report the reason of data transformation and detailed statistics procedure. We have therefore revised the “EEG data processing” subsection (page 5, lines 207–page 6, line 222) and “MEP data processing” subsection (page 6, lines 238–247) of the Materials and Methods as follows:

“In order to conduct a proper comparison for differences in power spectra between tACS frequency conditions, normality testing using Kolmogorov-Smirnov test was used. Based on the result of the Kolmogorov-Smirnov testing, either parametric two-way repeated measures analysis of variance (ANOVA) or nonparametric Friedman’s test was used. Additionally, for nonparametric testing, the logarithmically transformed power spectrum without normality distribution was normalized to baseline (i.e., before tACS) according to the following equation:

NP(f,t) = A(f,t)-R(f)/R(f),

(1)

where  denotes the normalized power spectrum,  represents the EEG power spectrum at time  and frequency  (i.e., the power spectrum of 10 and 20 Hz after tACS), and  denotes the mean power spectrum of the baseline period, defined as the 3-min interval before tACS. A large positive value indicates a large increase in the EEG power spectrum from the baseline period [46]. Furthermore, post-hoc analysis with parametric Bonferroni correction or the nonparametric Steel-Dwass test was performed to compare differences in power spectra between tACS frequency conditions. ”

“After state value estimation, normality testing using Kolmogorov-Smirnov test was used. Based on the result of the Kolmogorov-Smirnov testing, either parametric two-way repeated measures ANOVA or nonparametric Friedman’s test was used. For nonparametric testing, conditioned and unconditioned MEP amplitudes were normalized to the baseline data (Equation 1). In Equation 1,  denotes normalized MEPs,  denotes MEPs at time , and  denotes the mean MEP of the baseline period before tACS. A great positive value indicates a large increase in MEPs compared with that in the baseline period [46]. Post-hoc analysis with parametric Bonferroni correction or the nonparametric Steel-Dwass test was performed to compare differences in MEP amplitudes among the three tACS conditions.”

Comment 4

Again, I am confused why they applied a nonparametric test like the Friedman test. In the case of non-normality, they could transform data and then submit the values to an analysis of variance with time as within-participants factor and evaluate whether there was a main effect (e.g., a significant change between pre and post stimulation). Authors should better motivate their statistical choice. Also, in the results obtained with the Friedman test, I suggest reporting Chi squares and degrees of freedom.

Response to comment 4

We have revised the “Changes in brain oscillation and excitation” subsection (page 7, line 277 – page 9, line 326) of the Results as follows:

“The Kolmogorov-Smirnov test showed that the power spectra lacked normality (alpha oscillation before and after alpha-tACS, beta-tACS, and sham-tACS: both p < 0.0001). Therefore, nonparametric testing and Equation 1 was used for comparison of the power spectra of alpha and beta oscillations after alpha-, beta-, and sham-tACS treatments. The normalized power changes (i.e., the event-related synchronization [ERS] and event-related desynchronization [ERD]) in alpha oscillatory neural activities after alpha-, beta-, and sham-tACSs are shown in Figure 6A. The Friedman test showed a significant difference in power changes in alpha oscillations among alpha-, beta-, and sham-tACSs (chi-squared = 16.75, degree of freedom = 2, p = 0.0002). Additionally, post-hoc tests showed that alpha power oscillation was greater after alpha-tACS than after sham-tACS (alpha-tACS vs. beta-tACS: t = 1.37, p = 0.358; alpha-tACS vs. sham-tACS: t = 3.51, p = 0.001; beta-tACS vs. sham-tACS: t = 2.29, p = 0.057). Alpha power oscillation was greater after beta-tACS than after sham-tACS, but significance was not reached.

“The ERS/ERD of beta-oscillatory neural activities after alpha-, beta-, and sham-tACSs are shown in Figure 6B. The Friedman test showed a significant difference in the ERS/ERD of beta oscillations among the alpha-, beta-, and sham-tACSs (chi-squared = 11.53, degree of freedom = 2, p = 0.003). Additionally, post-hoc tests showed that beta power oscillation was greater after beta-tACS than after sham-tACS (alpha-tACS vs. beta-tACS: t = 1.65, p = 0.358; alpha-tACS vs. sham-tACS: t = 3.45, p < 0.0001; beta-tACS vs. sham-tACS: t = 4.34, p = 0.016). Moreover, beta power oscillation was lower after alpha-tACS than after sham-tACS.”

“The Kolmogorov-Smirnov test showed that the MEP amplitudes lacked normality (conditioned and unconditioned MEP before and after alpha-tACS, beta-tACS, and sham-tACS: both p < 0.0001). Therefore, nonparametric testing and Equation 1 was used for comparison of the conditioned and unconditioned MEP amplitudes after alpha-, beta-, and sham-tACS treatments. The changes in the normalized conditioned MEP amplitudes after tACS are shown in Figure 8A. The Friedman test showed a significant difference in normalized condition MEP amplitudes among alpha-, beta-, and sham-tACSs (chi-squared = 42.28, degree of freedom = 2, p < 0.0001). Further, post-hoc tests showed that the conditioned MEP amplitudes were smaller after alpha- and beta-tACS than after sham-tACS. Specifically, MEP amplitudes after alpha-tACS were smaller than those after beta-tACS (alpha-tACS vs. beta-tACS: t = 2.56, p = 0.029; alpha-tACS vs. sham-tACS: t = 4.93, p < 0.0001; beta-tACS vs. sham-tACS: t = 2.38, p = 0.045).”

“The normalized unconditioned MEP amplitudes changes after tACS are shown in Figure 8B. The Friedman test showed a significant difference among the alpha-, beta-, and sham-tACSs (chi-squared = 11.93, degree of freedom = 2, p = 0.002). Further, post-hoc tests showed that the unconditioned MEP amplitude were smaller after alpha- and beta-tACS than after sham-tACS (alpha-tACS vs. beta-tACS: t = 1.41, p = 0.338; alpha-tACS vs. sham-tACS: t = 3.80, p = 0.0004; beta-tACS vs. sham-tACS: t = 2.57, p = 0.028).”

Comment 5

In the Discussion section, page 10, line 15, Authors say: “ In addition, beta-tACS increases alpha oscillations as well”. This is not true based on the results reported, as the difference in alpha power after beta-tACS and after sham-tACS did not reach significance (p = 0.057). Here I suggest removing this sentence and limiting the discussion to the main findings, e.g., alpha- and beta-tACS modulated alpha and beta oscillations, respectively, and they differently influence cortical inhibition.

Response to comment 5

We have revised the Discussion (page 11, lines 373–374) as follows:

“Our experimental results showed that alpha- and beta-tACS result in larger alpha and beta oscillations, respectively, and they differently influence cortical inhibition.”
